# Laboratory layered latte

Nan Xue [1], Sepideh Khodaparast [1], Lailai Zhu [1,2], Janine K. Nunes[1], Hyoungsoo Kim [1,3] & Howard A. Stone [1]

Inducing thermal gradients in fluid systems with initial, well-defined density gradients results in the formation of distinct layered patterns, such as those observed in the ocean due to double-diffusive convection. In contrast, layered composite fluids are sometimes observed in confined systems of rather chaotic initial states, for example, lattes formed by pouring espresso into a glass of warm milk. Here, we report controlled experiments injecting a fluid into a miscible phase and show that, above a critical injection velocity, layering emerges over a time scale of minutes. We identify critical conditions to produce the layering, and relate the results quantitatively to double-diffusive convection. Based on this understanding, we show how to employ this single-step process to produce layered structures in soft materials, where the local elastic properties vary step-wise along the length of the material.

[1] Department of Mechanical and Aerospace Engineering, Princeton University, Princeton, NJ 08544, USA. [2] Linné Flow Centre and Swedish e-Science Research Centre (SeRC), KTH Mechanics, Stockholm, SE 10044, Sweden. [3] Department of Mechanical Engineering, Korea Advanced Institute of Science and Technology, Deajeon 34141, South Korea. Correspondence and requests for materials should be addressed to H.A.S. (email: hastone@princeton.edu)

Pattern forming systems are some of the intriguing and spectacular phenomena throughout science and technology[1–4]. In nature, patterns in fluid media, such as the waves on the surface of deep water[5, 6], oscillations in flames[7], large-scale von Kármán vortex streets in clouds[8], and the symmetric yet complex shape of snow flakes[9], constitute some of the earliest self-organized systems, which have attracted human curiosity and initiated scientific exploration. A considerable class of spatial patterns in fluids are structured due to thermal effects, which trigger hydrodynamic instabilities[10–12]. For example, well-known instabilities triggered by thermal effects, such as Rayleigh-Bérnard convection[13–15], are often found in systems with well-defined initial conditions.

In a fluid system, when thermal gradients are introduced in the presence of an initial well-defined density gradient, distinct layered patterns are observed similar to those sometimes found in the ocean due to double-diffusive convection[16–20]. Surprisingly, we observe distinct horizontal layers formed after haphazardly pouring espresso into a glass of warm milk. Pouring forces a lower-density liquid (espresso) into a higher-density ambient (milk). The downward liquid inertia caused by pouring is opposed by buoyancy. The dynamics is similar to the fountain effect[21, 22], which characterizes a wide range of flows driven by injecting a fluid into a second miscible phase of different density.

Here we perform controlled model experiments, injecting warm dyed water from the top into a cylindrical tank filled with warm salt solution. The mixture cools down at room temperature and multiple horizontal layers emerge over several minutes. We use light intensity in the digital images of the fluid in the tank, after the injection, to quantify the distribution of the mixture density. We show that the formation of horizontal layers is a result of double-diffusive convection, where the salinity and temperature gradients are applied vertically and horizontally, respectively. The presence of the circulating flows within the layers is confirmed via particle image velocimetry (PIV) experiments and numerical simulations. Furthermore, we report that the formation of the horizontal layers is controlled by the injection velocity, i.e. layers emerge only when the injection velocity is higher than a critical value. Finally, we propose a single-step procedure for fabricating multi-layer soft materials based on our understanding of the model system.

## Results

**Layered latte**. A glass of latte is made by pouring a cup of espresso into a glass of warm milk. Since the two liquids are miscible, the result of pouring is an espresso-milk mixture at the top of the glass, while the bottom may only contain milk, if no additional stirring is applied (Fig. 1a). In fact, although the initial state of the mixture is complex and chaotic (Fig. 1b), there are conditions where the mixture cools at room temperature and exhibits an organized layered pattern (Fig. 1c, see Supplementary Movie 1). These stable layers, whose structures may be maintained for at least tens of minutes (Fig. 1d), or even several hours, contain different concentrations of espresso and hence exhibit distinct visible boundaries.

**Experimental model**. In order to investigate the mechanisms leading to the layering of the mixture, we performed controlled experiments in a model system comprised of a low-density jet of dyed water (density $\rho_w = 0.992 \times 10^3$ kg m$^{-3}$ at $T_H = 40\,°C$, injection volume $V_I = 30$ ml) entering a tank filled with relatively higher-density brine (9.1 wt% sodium chloride solution, $\rho_s = 1.056 \times 10^3$ kg m$^{-3}$ at $T_H = 40\,°C$, 340.9 ml). The jet enters from the top (Fig. 1e) and the solution is then left to cool at room temperature $T_{atm} = 22\,°C$.

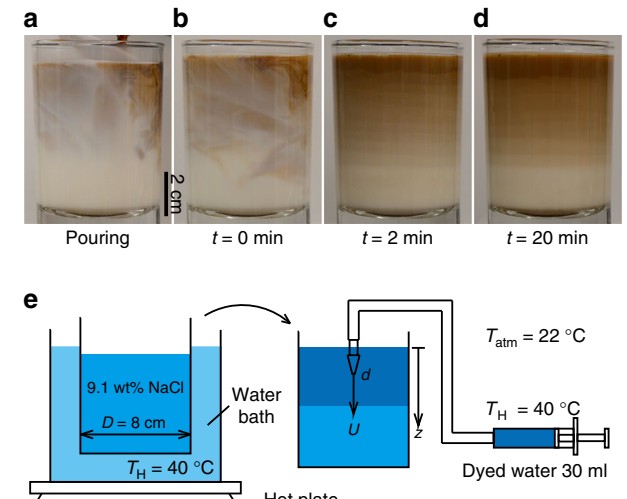

**Fig. 1** Pattern formation in an injection-driven system inspired by a layered latte. **a–d** Formation of horizontal layers in a glass of latte. **a** A small volume ($V_I = 30$ ml) of espresso (temperature $T_H = 50\,°C$) is poured into 150 ml of warm milk ($T_H = 50\,°C$). **b** Espresso and milk form a mixture, which exhibits chaotic dynamics caused by the injection. The resulting espresso-milk mixture remains at the top of the container due to buoyancy. **c** As the mixture cools down to room temperature, $T_{atm} = 22\,°C$, multiple horizontal layers of different espresso concentrations are formed. **d** These horizontal layers maintain their structures over time (see Supplementary Movie 1). **e** Schematic of model experiments. Dyed water (0.01 wt% methylene blue hydrate) and sodium chloride solution (9.1 wt%) are heated in a water bath at $T_H = 40\,°C$. Then the heated dyed water is injected vertically downward through a needle of diameter $d$ into the salt solution using a syringe pump, which allows controlling the flow rate and thus the injection velocity $U$ of the dyed water. Finally, all of the liquid in the tank cools down at room temperature and layers are observed. The cooling begins as soon as the injection starts and lasts for at least several minutes

**Experimental observations**. When the dyed water is injected into the higher-density salt solution, a downward jet is generated. However, the penetration of this liquid jet into the salt solution is opposed by the buoyant force pushing the lower-density liquid jet back to the top of the tank (Fig. 2a, c). As a result, a mixture is formed, in which the dyed water is mixed with the salt solution and is separated from the original salt solution at the bottom of the tank. In addition to buoyancy, the mixing is mainly governed by inertia with the Reynolds number defined as Re = $Ud/2\nu \approx O(100)$ ($U$ is the injection velocity, $d$ is the diameter of the needle and $\nu$ is the kinematic viscosity of the fluid), while diffusion does not play a significant role during the injection. The Schmidt number is Sc = $\nu/\kappa_s \approx 300$ ($\kappa_s$ is the mass diffusivity of the salt), indicating that the momentum diffusion is far faster than the salt diffusion during both the injection and layering (if any) processes. The Schmidt number for the milk and espresso system is approximately $10^4 \gg 1$, and so the system is similar to the model salt and water system, in which momentum diffusion dominates. At relatively low injection velocities (Fig. 2a, b), the mixture of dyed water and salt solution (initially at $T_H = 40\,°C$) remains unchanged as it cools down at room temperature. However, above a critical injection velocity, multiple layers similar to those observed in the glass of latte (Fig. 1a–d) are formed in the mixture several minutes after the injection (Fig. 2d). Once formed, the layers are not influenced by external mechanical disturbances, and can recover even after gentle stirring. As the cooling continues, these layers may merge and form thicker structures, which can last for days before being entirely eliminated by

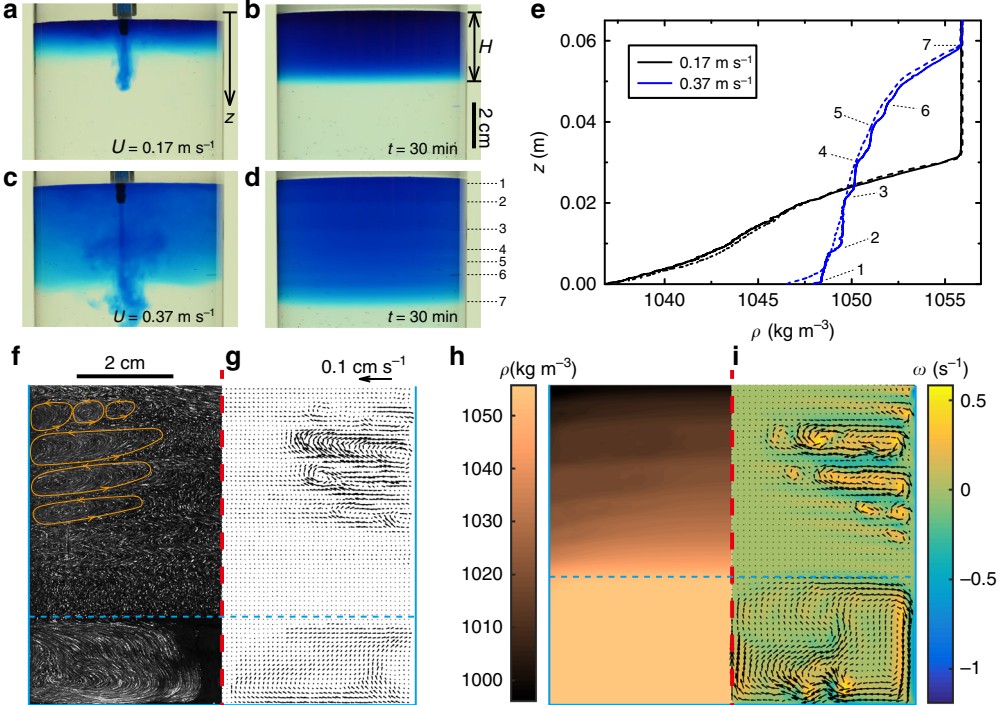

**Fig. 2** Evidence for double-diffusive convection in the model experiments and simulations. **a–d** Dyed water (0.01 wt% methylene blue hydrate, 30.0 ml) is injected into 9.1 wt% sodium chloride solution (340.9 ml) with two different injection velocities: $U = 0.17\,\mathrm{m\,s^{-1}}$ (**a**, **b**) and $U = 0.37\,\mathrm{m\,s^{-1}}$ (**c**, **d**). **a**, **c** correspond to the dynamics during the injection and (**b**, **d**) to the mixture 30 min later. The distinguished boundaries of the layers are marked and enumerated. **e** The density profiles of the mixtures presented in **a–d**. The dashed lines refer to the density profiles 1 min after the injection, and the solid lines refer to those observed 30 min later. The black and the blue lines refer to the results at injection velocities $U = 0.17\,\mathrm{m\,s^{-1}}$ and $U = 0.37\,\mathrm{m\,s^{-1}}$, respectively. Vertical steps (solid blue line) indicate layers in the mixture, and their boundaries are enumerated according to those in **d**, where $z = 0$ refers to the top of the mixture. **f**, **g** Particle streak-lines (**f**) and the velocity vectors (**g**) measured in PIV experiments for $U = 0.37\,\mathrm{m\,s^{-1}}$, 10 min after the injection. The vertical dashed line refers to the axis of the cylindrical domain, and the horizontal dashed line marks the bottom of the mixture. **h**, **i** Numerical results (details in Methods, Supplementary Figs 3 and 4 and Supplementary Discussion) show the density distribution (**h**) as well as the velocity vectors and the vorticity magnitude $\omega$ (**i**), 10 minutes after the injection with $U = 0.37\,\mathrm{m\,s^{-1}}$. The scales for the length and the velocity magnitude are identical in **g** and **i**

diffusion (Supplementary Fig. 1). The layers can be observed in the milk and espresso or in the salt and water mixture, only when the initial temperature is different from the room temperature.

We quantify the concentration of the salt in the mixture using the concentration of the blue dye as an indicator (Supplementary Fig. 2). Therefore, the local intensity of the blue dye in the digital images (Fig. 2a–d) can be correlated with the local density of the mixture. The dashed lines in Fig. 2e represent the initial density profiles for two different injection speeds, while the solid lines refer to the density profiles 30 min later. For both experiments the dashed lines exhibit continuous monotonically increasing density profiles, when moving from the top to the bottom of the mixture. While the density profiles before and after the injection at a low velocity $U = 0.17\,\mathrm{m\,s^{-1}}$ (black) remain almost identical, the density profile 30 min after the injection at a higher velocity $U = 0.37\,\mathrm{m\,s^{-1}}$ (blue) exhibits clear steps indicating the formation of horizontal layers. After the high-velocity injection, the density in a single layer is constant, implying that the liquid within a layer is uniformly mixed. Moreover, the discontinuities in the density profile are clearer near the top of the mixture, where the gradient of the density is smaller than that of the bottom layers. We postulate that this layered state is reached due to the double-diffusive convection, which is well known in layer formation in open water systems such as oceans or lakes[18–20, 23, 24]. In our experiment, the double-diffusive convection results from the combination of heat transfer to the surroundings from the warm

liquid, and a density gradient generated in the mixture from the initial pouring or injection.

A directional heat transfer phenomenon in a mixture with an initial density gradient has been observed previously in other systems to lead to the formation of well-defined layers in fluid mixtures due to double-diffusive convection[18–20, 23, 24]. In particular, when a given temperature difference is created between two vertical walls bounding a fluid with an initial vertical density gradient, the fluid near the cold wall is cooled and thus gets denser and sinks. The sinking of the liquid due to the heat transfer will however be suppressed as the cooler liquid close to the wall reaches a zone of a similar density in the mixture. Therefore, the downward-moving liquid starts flowing inwards away from the cold wall as it can no longer proceed in the vertical direction. A similar motion but in the opposite sense is created close to the warmer wall. Consequently, closed streamlines are formed in the fluid circulating between the cold and the warm sources[23]. Within the circulation cells, the fluid mixes uniformly, and thus the density is fairly constant, while each circulation cell acquires a different density.

In order to verify the postulate of double-diffusive convection as the cause of layering in our confined injection-driven system, we performed experimental (Fig. 2f, g) and numerical analyses (Fig. 2h, i) of the time-dependent flows (see details in Methods, Supplementary Figs 3, 4 and Supplementary Discussion). Both approaches document the formation of recirculating patterns in the form of axisymmetric rings between the wall and the center of

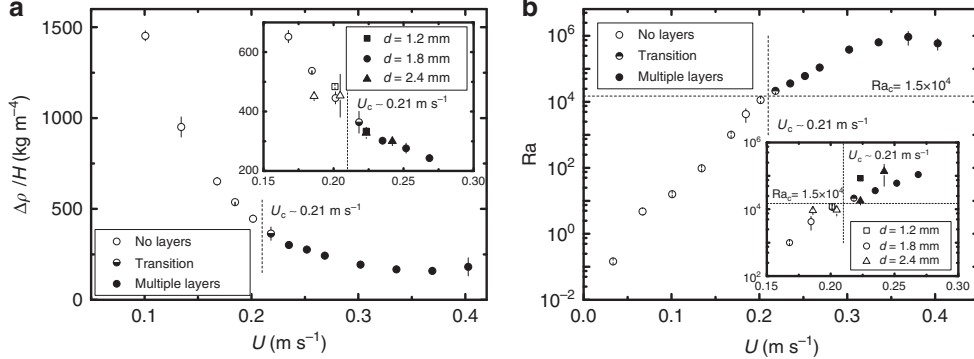

**Fig. 3** Role of the injection velocity $U$. **a** The average density gradient, defined as the density difference between the bottom and the top of the mixture relative to the mixture height $H$, as a function of $U$. **b** The Rayleigh number Ra, defined based on the minimum density gradient and the maximum temperature difference in the mixture, vs. $U$. The hollow, half-solid and solid symbols refer to the observations, where, respectively, multiple layers are never, sometimes and always observed in the mixture. The squares, circles and triangles refer to different inner diameters of the injection needles $d = 1.2$, 1.8 and 2.4 mm, respectively. The insets show a zoomed-in view around the transition region. The horizontal dashed line in (**b**) at $Ra_c = 1.5 \times 10^4$ indicates the critical Rayleigh number reported in the literature[18]. Error bars correspond to standard derivations with $n$ experiments with $n = 3 - 5$

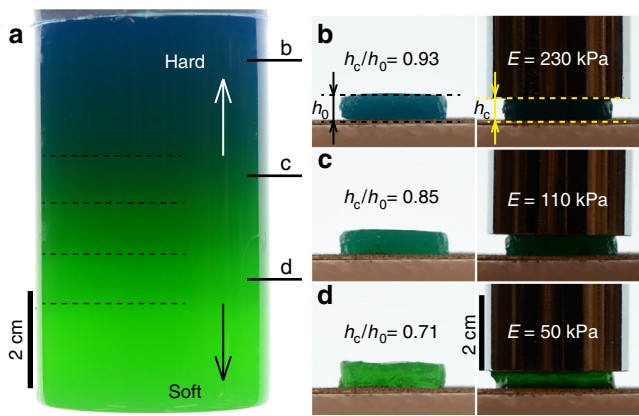

**Fig. 4** Single-step fabrication of soft layered gels. **a** Horizontal layers (in different colors) are created from an agarose solution in a single-step process by injecting 100 ml 4 wt% agarose solution with 0.0015 wt% methylene blue hydrate into 100 ml 9.1 wt% NaCl solution containing 0.002 wt% fluorescein sodium salt. Boundaries of the layers are marked with dashed lines. The layers differ from one another in softness due to the different concentrations of agarose, i.e. the bottom layers are softer than those at the top. **b**–**d** Cylindrical gel slabs, which are cut from different layers in the gel, show different mechanical properties. Therefore, applying the same vertical load on the slabs results in different degrees of compression and thus indicates the different Young's moduli of the gel in these horizontal layers

the tank. The boundaries separating these circulation cells overlap with the limits of the layers in the mixture.

**Critical injection velocity**. The circulation cells are the results of the competition between the horizontal thermal gradient and the vertical density gradient generated by the fluid injection, i.e., the thermal gradient triggers the fluid to rise or sink close to the boundaries, while the vertical density gradient opposes this motion and subsequently stabilizes the flow. In our model experiments, layers are observed only above a critical injection velocity $U_c$ (Fig. 2a–d and Supplementary Fig. 5). At higher injection velocities, the depth of penetration of the liquid jet increases, and similarly the thickness of the mixture $H$ increases, indicating that the dyed water mixes with a larger volume of salt solution. For a fixed volume of injected water ($V_I = 30$ ml), we found that the volume of the resulting mixture $V_M$ increases

linearly with the injection velocity $U$ for the range of parameters studied here. Consequently, the mixing level and the density gradient in the mixture of volume $V_M$ are controlled by the injection velocity.

We define an average density gradient $\Delta\rho/H$ in the mixture, where $\Delta\rho$ is the magnitude of the density difference between the bottom and top of the mixture. Also, $\Delta\rho/H$ indicates the average resistance imposed by the salinity gradient against the formation of a convection cell due to the thermal gradient and is a function of $U$. Our experimental measurements show that $\Delta\rho/H$ decreases with increasing $U$ due to enhanced mixing (Fig. 3a). We find that for $U > U_c \approx 0.21 \, \text{m s}^{-1}$ the resistance from salinity can no longer compete with the thermal gradient, thus circulation cells appear and multiple layers emerge.

The competition between the thermal cooling and the salinity gradient in this double-diffusive convection, where the horizontal temperature gradient is orthogonal to the vertical salinity gradient, is characterized by the Rayleigh number, $\text{Ra} = \frac{g\alpha\Delta T}{\nu\kappa}\left[\frac{\alpha\Delta T}{(-\text{d}\rho/\text{d}z)/\rho_w}\right]^3$, where $g$ is the gravitational acceleration, $\alpha$ is the coefficient of fluid volume expansion, $\Delta T$ is the temperature difference in the fluid, $\nu$ is the kinematic viscosity of the fluid, $\kappa$ is the thermal diffusivity of the fluid and $(-\text{d}\rho/\text{d}z)/\rho_w$ is the normalized salinity gradient in the mixture[18]. In a system with an initial linear salinity gradient and an imposed horizontal temperature gradient, the critical Rayleigh number $\text{Ra}_c$ for initiation of an instability and formation of the layers has been found to be around $\text{Ra}_c \approx 1.5 \times 10^4$ experimentally[18]. Note that in a conventional double-diffusive convection problem, a linear density gradient is imposed at the initial stage and the temperature gradient is created between two vertical bounding walls by setting different constant temperatures[18, 24]. The initial and boundary conditions leading to the layers in a glass of latte and our model experiments are, however, different from those in the traditional problem: (1) the density gradient is caused by the injection and is not constant in the mixture (Fig. 2e) and (2) the temperature gradient is not constant during the experiments as the core of the liquid cools down continuously. Therefore, to characterize our system and to calculate Ra, we chose to use the minimum density gradient over the typical thickness of the layers (around 5 mm) at the corresponding injection velocity and the maximum temperature gradient. We found that the critical Rayleigh number in our experiments closely matches the value reported in the literature for an idealized configuration $\text{Ra}_c \approx 10^4$ (Fig. 3b). Consistent with our observations, $\text{Ra}_c$ indicates that the

layering is obtained only when $U$ is above $U_c$. Consideration of the dynamics in a proper dimensionless framework requires an analysis with at least both the Froude and Reynolds numbers, which is a topic of on-going research. In addition, when layering occurs, the expected length scale (thickness) of a layer is approximately $(\alpha \Delta T)/[(-\mathrm{d}\rho/\mathrm{d}z)/\rho_w]$.

**Application to soft materials**. The double-diffusive convection and the formation of the layers are simply controlled by the thermal and salinity gradients in the fluid systems discussed above, which implies no conceptual restriction on applying this principle to more complex fluid systems, such as thermally established soft gels. There are various approaches for generating layered soft materials, but most of these approaches are currently multi-step processes, where solid layers are usually formed sequentially[25]. Nevertheless, based on the understanding outlined above, we can make multiple layers in soft materials (UltraPure™ Agarose, Invitrogen) simply by a single step of injecting ($U \approx 1\,\mathrm{m\,s^{-1}} > U_c$) a hot gel solution into a denser solvent and cooling the mixture at room temperature (Fig. 4, see the gel recipe in Supplementary Methods). To further demonstrate the presence of the horizontal layers in the gel, we performed experiments with the same recipe but measured the light intensities in the digital images of the gel rather than the elastic properties (see Supplementary Fig. 6 and Supplementary Discussion). While cooling, horizontal layers are first formed in the agarose solution, which is subsequently solidified to a layered gel below the gelation temperature. The Young's moduli in the final layered state formed from the same agarose solution vary systematically with the vertical position (50 kPa at the bottom compared to 230 kPa at the top), which implies that the concentration of agarose in distinct layers is different. Further, the difference of concentration in the gel layers leads to a difference of porosity in these layers, so the concentration and the diffusion rate of additives will be different. This single-step, single-chemistry method can facilitate the fabrication of multiple-layered structures in food science[25], tissue engineering[26, 27], and other applications in materials science.

## Methods

**Model experiment setup**. In our model experiments with salt solutions, blue dye (methylene blue hydrate, Sigma-Aldrich, 0.01 wt%) is added to the water jet to visualize the mixing of the two liquids. Dyed water is injected downwards using a syringe pump (Harvard Apparatus PHD 2000) through nozzles of circular cross-section, with inner diameters $d = 1.2, 1.8, 2.4$ mm, into a polystyrene cylindrical tank with diameter $D = 8$ cm (Fig. 1e). Working liquids are brought to the final temperature ($T_H = 40 \pm 1\,^{\circ}$C) in a water bath. We controlled the flow rate of the injected water and consequently the inlet velocity $U$ using the syringe pump. After the injection, the tank is cooled down at room temperature. The appearance and evolution of the layers are visualized by placing the tank in between a LED panel and a camera, while the flow velocities in the mixture are obtained by following tracer particles in the PIV experiments.

**Density profile in the mixture**. We performed a calibration procedure to correlate the local intensity of the blue dye in the digital images to the local concentration $c$ (mass ratio) of injected dyed water containing 0.01 wt% methylene blue hydrate (see Supplementary Fig. 2). Therefore, we calibrated the local intensity of the blue dye to obtain the local mass ratio of injected dyed water in the mixture, and then calculated the local density in the mixture by considering $\rho = c\rho_w + (1 - c)\rho_s$, where $\rho_w$ is the density of water and $\rho_s$ is the density of the salt water initially in the tank.

**Flow visualization and PIV**. The water injected from the top of the tank is labelled with blue dye; therefore the concentration of the dye indicates the amount of water and consequently the salt concentration in the mixture as the blue jet mixes with the salt solution in the tank. The depth of the mixture and the layers formed at higher flow rates are determined by placing the tank in front of a large LED panel and capturing color images of the mixtures over a long period of time (up to 3 days, see Supplementary Fig. 1).

In order to quantitatively visualize the structure of the flow in the mixture, the liquid in the tank is seeded with tracer particles (PSP-20, diameter 20 μm, Dantec Dynamics). The plane of symmetry in the cylindrical tank is illuminated with a green light sheet (thickness of ≈1 mm) created by placing a laser line lens (PL0160, Thorlabs) in front of a green laser (BioRay 520, Coherent). Images are captured using a DSLR camera and a macro lens at the rate of 30 frames per second. The standard deviation of the light intensity for each pixel in the sequence of the recorded grey-scale images is calculated to determine the path lines of the particles in the mixture (Fig. 2f). Moreover, the ensemble cross-correlation scheme is applied to the sequence of grey-scale images to measure the local velocities in the PIV analyses[28]. Square interrogation windows of $32 \times 32$ pixels corresponding to grid cells of $1 \times 1$ mm$^2$ with an overlap of 50% are used to obtain the velocity vectors, such as those presented in Fig. 2g.

**Numerical simulations**. We consider double-diffusive convection of an incompressible flow of a Newtonian fluid inside a cylindrical container (after the injection). The density $\rho$ of the fluid varies with the temperature $T$ and the salinity $S$ following the Boussinesq approximation

$$\rho = \rho_0[1 + \beta(S - S_0) - \alpha(T - T_0)], \quad (1)$$

where $\rho_0$, $S_0$ and $T_0$ denote, respectively, the density, salinity and temperature of the reference state and $\beta$ (respectively $\alpha$) indicates the solutal (respectively thermal) expansion coefficient. In the case of small-density variations as in our case, this approximation is well justified. Other parameters of the problem include the thermal diffusivity $\kappa$, kinematic viscosity $\nu$ of water, solutal diffusivity $\kappa_s$ and gravitational acceleration $g$.

We choose the radius $R$ of the cylinder, $\kappa/R$ and $\rho\kappa^2/R^2$ to scale the length, velocity and pressure, respectively. We introduce the non-dimensional temperature $\theta$ and salinity $\sigma$ as

$$\theta = (T - T_{\max})/(T_{\max} - T_{\min}), \quad (2)$$

$$\sigma = (S - S_{\max})/(S_{\max} - S_{\min}), \quad (3)$$

where $T_{\max/\min}$ represents the maximum and minimum temperature and likewise for $S_{\max/\min}$. The non-dimensional equations have the form

$$\nabla \cdot \mathbf{U} = \mathbf{0}, \quad (4)$$

$$\frac{\partial \mathbf{U}}{\partial \tau} + \mathbf{U} \cdot \nabla\mathbf{U} = -\nabla P + \mathrm{Pr}\nabla^2\mathbf{U} + \mathrm{Ra_T Pr}(\theta - N\sigma)\mathbf{e_z}, \quad (5)$$

$$\frac{\partial \theta}{\partial \tau} + \mathbf{U} \cdot \nabla\theta = \nabla^2\theta, \quad (6)$$

$$\frac{\partial \sigma}{\partial \tau} + \mathbf{U} \cdot \nabla\sigma = \frac{1}{\mathrm{Le}}\nabla^2\sigma, \quad (7)$$

where $\mathbf{U}$, $P$ and $\tau$ are the non-dimensional velocity, pressure and time, respectively; $\mathrm{Pr} = \nu/\kappa$ is the Prandtl number, $\mathrm{Ra_T} = g\alpha(T_{\max} - T_{\min})R^3/(\kappa\nu)$ the thermal Rayleigh number, $\mathrm{Le} = \kappa/\kappa_s$ the Lewis number and $N = \alpha(T_{\max} - T_{\min})/\beta(S_{\max} - S_{\min})$ indicates the buoyancy ratio.

We solve the governing equations Eqs. 4–7 in the $(r, z)$ cylindrical coordinates by employing the commercial finite element solver COMSOL. The assumption of azimuthal independence is verified a posteriori by comparing the numerical and experimental results. We use approximately 6000 quadrilateral elements (validated with 30,000 quadrilateral elements) to discretize the computational domain, and the near-wall mesh is carefully refined in order to resolve the thermal boundary layers. Quadratic elements are adopted for $(\mathbf{U}, \theta, \sigma)$ and linear elements for $P$. It is worth noting that any options for numerical diffusion in the COMSOL's CFD module have been deactivated.

We now describe the boundary conditions (BCs). They are illustrated in the sketch of the computational domain consisting of four boundaries: the axis (left), walls (right and bottom), and the free-slip surface (top) (see Supplementary Fig. 3). On the two walls we impose the no-slip BCs $U_r = U_z = 0$, on the axis $U_r = 0$, and on the top surface we apply zero normal velocity $U_z = 0$ and zero tangential stress $(\mathbf{I} - \mathbf{nn}) \cdot [(\nabla\mathbf{U} + (\nabla\mathbf{U})^\mathsf{T}) \cdot \mathbf{n}] = \mathbf{0}$, where $\mathbf{n}$ is the outward-pointing normal vector. Zero-flux $\mathbf{n} \cdot \nabla\sigma = 0$ is imposed for the salinity $\sigma$ on all the boundaries. The same condition applies for the temperature $\theta$ except for the right wall, which is modeled as a conductive BC transferring the heat inside the container towards the ambient air due to the temperature difference. The conductive BC reads $\mathbf{n} \cdot \nabla\theta = \mathrm{Nu}(\theta_{\mathrm{atm}} - \theta)$, where $\theta_{\mathrm{atm}}$ is the non-dimensional air temperature; also, $\mathrm{Nu} = hR/k$ denotes the Nusselt number, where $h$ and $k$ correspond to the heat transfer coefficient and heat conductivity. Finally, as the initial condition, we choose the initial density profile measured in the experiments after mixing by the injection.

Our implementation has been validated against ref. [29] and our 2D planar version against ref. [30]. The reader is also referred to ref. [31] for other flow cases where the results of COMSOL simulations show excellent agreement with the asymptotic analysis.

**Calculating the Rayleigh number**. We calculated the Rayleigh number $Ra = \frac{g\alpha\Delta T}{\nu\kappa}\left[\frac{\alpha\Delta T}{(-d\rho/dz)/\rho_w}\right]^3$ for the stability of thermal convection in a salinity gradient due to lateral heating[18]. In our calculation, the gravitational acceleration is $g = 9.8\,\mathrm{m\,s^{-2}}$, while the kinematic viscosity and the thermal diffusivity of fluid are $\nu = 6.6 \times 10^{-7}\,\mathrm{m^2\,s^{-1}}$ and $\kappa = 1.5 \times 10^{-7}\,\mathrm{m^2\,s^{-1}}$, respectively at $T_H = 40\,^\circ\mathrm{C}$. The coefficient of fluid volume expansion is $\alpha = 3.9 \times 10^{-4}\,\mathrm{K^{-1}}$ at $T_H = 40\,^\circ\mathrm{C}$. The temperature difference in the fluid is $\Delta T = 2\,^\circ\mathrm{C}$, which is the maximum temperature difference measured (by attaching thermocouples) between the center and the wall of the container during cooling at room temperature. Also, $(-d\rho/dz)$ is the minimum of the local salinity gradient in the mixture and varies with the injection velocity $U$. We calculated the local slopes of the density gradient curves (Fig. 2b), over a height of $\delta z = 5$ mm, which represents the minimum thickness of layers that we observed in our experiments. We divided this local slope by the density of water to obtain the local salinity gradient in the mixture $(-d\rho/dz)/\rho_w$. The minimum value of $(-d\rho/dz)/\rho_w$ in the mixture is used to calculate the Rayleigh number (Fig. 3b).

**Data availability**. The datasets generated during and analyzed during the current study are available at http://github.com/xuenan1203/Laboratory-Layered-Latte.

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

## Acknowledgements

S.K. thanks the Swiss National Science Foundation (P2ELP2-158896) for funding. L.Z. thanks the Swedish Research Council (2015-06334) for a VR International Postdoc Grant. J.K.N. and H.A.S. thank the National Science Foundation (CMMI-1661672) for partial support. We thank Jie Feng, Y. Estella Yu, Suin Shim and Antonio Perazzo for valuable discussions, Ching-Yao Lai for suggestions on gel preparation and Bob Fankhauser for providing an interesting picture of layered patterns formed in coffee that inspired this work.

## Author contributions

N.X. and H.A.S. initiated this work. N.X., S.K. and J.K.N. performed the experiments. L. Z. performed the numerical simulations. N.X., S.K., L.Z., J.K.N., H.K. and H.A.S. contributed to analyzing the data, discussing the results and writing and revising the paper.
