## [Peer Review File · Nature Communications]

Reviewers' comments:

Reviewer #1 (Remarks to the Author):

The paper describes beautiful and controlled experiments on pouring espresso into milk, observing layer formation as the mixture cools down from the side. Next to this systems known by many, even more controlled experiments are done by system of salt solution and dyed water, from which most data are taken.

The crucial control parameter is the pouring velocity, with which the light liquid is forced into the heavy one. Only beyond a critical pouring velocity does the layer formation occur. The system is identified as double diffusive convection (DDC), with the Rayleigh number of the destabilizing field as the crucial control parameter. The experiments are supplemented by COMSOL simulations.

I like the findings of the paper and the paper itself very much. It is very original, nicely and carefully done, very relevant for applications in material science (as the last part of the paper shows, see figure 4). Clearly, it will attract a lot of attention.

However, before the paper can be published, the following changes and additions should be done:

1. Ref [13-17] on DDC and pattern formation in RB are all very classical papers, many of them decades old. I strongly suggest to also give some more recent literature, e.g. the nice book by T. Radko on DDC (Cambridge Univ. Press) and the ARFM by Bodenschatz, Pesch, and Ahlers on pattern formation in RB (2000).
2. Clearly, the Schmidt number of the system should be given and at least be estimated. I guess it is rather larger, say, at least several hundreds. This also allows putting the system into context of DDC in the ocean or in lakes and the well-known layer formation there. Clearly, that phenomenon should be mentioned, see the Radko book. Calculating the Schmidt numbers will also show how similar the model system (salt-water) is to the milk-express system.
3. DDC is controlled by TWO Rayleigh numbers, namely the one for the temperature and the one for the concentration field. They can be either positive (destabilizing) or negative (stabilizing). Both Rayleigh numbers should be given.

4. One reason why I worry on the Schmidt number are the numerical simulations by COMSOL. I somehow doubt that COMSOL can handle so large Schmidt numbers as it would require excellent grid resolution for the concentration field (which has much smaller diffusivity). Anyhow, the authors say that they have validated their code and found a good match, but clearly this remains some major concern. Perhaps it still works as the Rayleigh numbers are not high.

5. The numerical simulations are 2D in the sense that (r,z) cylindrical coordinates are used. Why so? Why not simply 2D in the (x,z) plane? Note that there are considerable difference between 2D and 3D RB convection. The best of course would be 3D DDC, but indeed I can image that COMSOL then fully fails.

6. Figure 2b: The authors may consider to switch the axis and plot z vs ρ , as more common in DDC. But it is of course up to them.

Reviewer #2 (Remarks to the Author):

This paper presents an experimental and numerical examination of an elegant method for producing layering using a simple double diffusive fluid system. Layering is of relevance to a number of industrial and environmental processes and so the identification of techniques to simply and repeatably reproduce layering are worthy of publication in their own right. The study presents some details of the physics behind the layering process which provides insight into the underlying processes involved. The study then goes on to demonstrate application of their technique for producing layers by application to soft matter gels which are of wide use in modern society.

The paper is very well written and well presented. I would not be unhappy to see the paper published as is but I have four comments which the authors may like to consider:

1) The layers in the relevant images are visible when I view on-screen but in my printed version they are impossible to detect. I wonder if any more can be done, e.g. some carefully targeted stretching of the light intensity values, to the images to accentuate the visible appearance (or otherwise) of the layers in the images?

2) I dislike the notion that the formation of layers is dependant on solely the velocity of injection. The appearance of layers must in general result from the initial stratification established and the rate of heat transfer through the walls of the vessel. The initial stratification must depend on the velocity of the injection but only in combination with the opposing effects of buoyancy, which typically for these fountain-like flows is expressed via a Froude number. Indeed, considerations in this vein provide that for sufficiently forced injections, the volume of 'mixed fluid' would be expected to vary linearly with injection volume (as is described to have been observed on p7) since the bulk mixing/entrainment by sufficiently forced fountain is linear in the Froude number (see for example Burrige HC, Hunt GR, 2016, Entrainment by turbulent fountains, Journal of Fluid Mechanics). I accept that you did not vary the Froude number except by varying the velocity but I think you should at least comment that simple considerations of the physics allows one to identify that the mixing process must be controlled by the Froude number of the injection. Personally I would present the horizontal axis in figure 3b in terms of a Froude number as the information regarding the critical velocity, only relevant to your particular set-up, is in figure 3a already – information regarding a critical Froude number may be applied to experiments of different initial density differences at the very least. Finally on this point, I would like to see a comment that one might expect the height of the layers to be determined by the initial density gradient and rate of change of fluid density due to heat transfer through the vessels side-walls, i.e. the characteristic distance 'cool' fluid falls before being neutrally buoyant in the stratified 'mixed' fluid. I would imagine that the layers only form when the height scale of the layers is large compared with a length associated with diffusion within the fluid but small compared with the height of the initial mixed region.

3) The jump, on page 8, from the discussion of critical Rayleigh numbers to the application of the technique to soft matter gels is far too stark and detracts from this interesting final application. There need to be a(some) more introductory sentence(s) beginning the the section on soft matter gels.

4) My only concern that truly requires addressing prior to publication, is that the image presented in figure 4a bears little evidence of layering (both when viewed on-screen and in-print). For example there is no indication that between any two consecutive dashed horizontal lines drawn on figure 4a the gel is of a constant colour – quite the opposite, if one ignores the horizontal dashed lines then my eyes (at least) identify a relatively continuous variation in colour over the entire region in which the colour is varying. If the variation is continuous then one would still expect the material properties to vary with height, which you demonstrate but this is not evidence of genuine layering. The real question is: if you divided the entire material in figure 4a into sections, each of a height equal to half the distance between your dashed lines, then could one identify a significant and systematic variation in the material properties based on whether two consecutive samples lay either within the same two dashed horizontal lines or were separated by a dashed horizontal line? Addressing this point is key to being able to claim genuine layering, rather than a continuous

variation, in the material properties of the gel formed. If more robust evidence cannot be produced then at the very least this points needs to clearly discussed and acknowledged in the text.

Reviewer #3 (Remarks to the Author):

The manuscript describes the formation of density layers in a cooling mixture of brine and dyed water using a jet of opposing buoyancy. This phenomenon is also observed while injecting coffee in milk and is rationalized as a thermosolutal convection process where the diffusivities of heat and solute produce density differences that drive fluid motion. The authors show in their model experiments and numerical simulations that the formation of layers is due to recirculating flow cells, which locally mix liquids. A critical velocity for the onset of fluid layering is reported and a similar approach is employed for generating layered soft materials. The manuscript presents interesting results but requires a few clarifications, including:

(1) The formation of horizontal layers is evidenced though analysis of grey-scale intensity profiles as explained in the SI. Data displayed in Fig. 2(b) of the main manuscript, however, do not provide compelling evidence of fluid layering as density steps only appear in a small portion of the graph and for one case. A revised figure clearly showing multiple step-wise intensity profiles would be more convincing. Such clarification is also necessary for Fig. 2 of the SI as intensity layers also appear in the images associated with low injection velocities (i.e., $U = 0.07$ and 0.20 m/s) where no layering is reported. In addition, a few quantitative remarks about the widths of the layers observed in both experiments and simulations would allow for a better connection between these approaches.

(2) Although the presence of a concentration gradient is apparent during the formation of the soft material in Fig. 4, no evidence is provided for the formation of a layered structure with a step-wise variation of the Young modulus E . A clear quantitative distinction should be made between a smooth and a step-wise variation of material properties. Providing the step-wise evolution of E as a function of z measured from consecutive thin slices would better substantiate the formation of layers resulting from double-diffusive convection.

(3) It has been reported that double-diffusive phenomena can also occur in the absence of heat when two solutes, such as salt and sugar, are present [Ref. 14, page 47]. In the reported experiments, salt and dye also constitute two different solutes and it would be useful to clarify whether additional experiments with variations of salt and dye concentrations along with various temperature differences were conducted. In general, more details about the influence of temperature would strengthen the manuscript.

(4) The schematic of the model experiment presented in Fig. 1(b) suggests at first that step (2) and (3) are conducted at the same time.

ANSWERS TO REFEREE 1

Referee: The paper describes beautiful and controlled experiments on pouring espresso into milk, observing layer formation as the mixture cools down from the side. Next to this systems known by many, even more controlled experiments are done by system of salt solution and dyed water, from which most data are taken. The crucial control parameter is the pouring velocity, with which the light liquid is forced into the heavy one. Only beyond a critical pouring velocity does the layer formation occur. The system is identified as double diffusive convection (DDC), with the Rayleigh number of the destabilizing field as the crucial control parameter. The experiments are supplemented by COMSOL simulations.

I like the findings of the paper and the paper itself very much. It is very original, nicely and carefully done, very relevant for applications in material science (as the last part of the paper shows, see figure 4). Clearly, it will attract a lot of attention.

However, before the paper can be published, the following changes and additions should be done.

Our response.

We thank the referee for his/her valuable comments. We are pleased to read “I like the findings of the paper and the paper itself very much. It is very original, nicely and carefully done, very relevant for applications in material science. Clearly, it will attract a lot of attention.”

Below we address the comments and suggestions of the referee and include the modifications to the manuscript wherever applicable.

1) Referee: Ref [13-17] on DDC and pattern formation in RB are all very classical papers, many of them decades old. I strongly suggest to also give some more recent literature, e.g. the nice book by T. Radko on DDC (Cambridge Univ. Press) and the ARFM by Bodenschatz, Pesch, and Ahlers on pattern formation in RB (2000).

Our response.

We thank the referee for this suggestion. Both references are cited in appropriate locations in the revised manuscript. We have also added additional reference to double diffusive convection that have come to our attention as we have prepared this revision.

Modification in the revised manuscript.

References are cited in the abstract and the first paragraph of the introduction in the revised manuscript, as Ref. [14] and [20]. We also cited a recent paper on double diffusive convection by Y. Yang, R. Verzicco and D. Lohse (2016) in the first paragraph of the introduction, as Ref. [15] (in the revised manuscript).

2) Referee: Clearly, the Schmidt number of the system should be given and at least be estimated. I guess it is rather larger, say, at least several hundreds. This also allows putting the system into context of DDC in the ocean or in lakes and the well-known layer formation there. Clearly, that phenomenon should be mentioned, see the Radko book. Calculating the Schmidt numbers will also show how similar the model system (salt-water) is to the milk-express system.

Our response. We thank the referee for the suggestion on the estimation of the Schmidt number in our system. We estimated the Schmidt number of our model experiments and it is approximately 300, as expected. As the Schmidt number (Sc) is a dimensionless number defined as the ratio of momentum diffusivity (kinematic viscosity) to mass diffusivity, this large Schmidt number indicates that the momentum diffusion is far faster than the mass diffusion. This conclusion agrees well with our experimental observations, which show that the diffusion of the mass (salt) occurs over days within the mixture, whilst the layers (a result of convection) are formed in few minutes.

We mentioned the double diffusion convections and layering in the oceans and lakes in the introduction of the paper, and we have now emphasized more about that phenomenon in open water, since the sodium chloride solutions used in our model experiments are similar to open water system.

Moreover, we estimated the Schmidt number for the milk and espresso system. The kinematic viscosity of the milk is approximately 2×10^{-6} m²/s and the mass diffusivity of milk varies from 0.24 to 2.1×10^{-10} m²/s (D. Doulia, K. Tzia, V. Gekas, Int. J. Food Prop., 2000). As a result, the estimated Schmidt number should be $> 10^4$. Such large Schmidt numbers indicate the momentum diffusion is much far faster than the mass diffusion and is similar to the salt and water system since the Schmidt number is rather high. In addition, the similar layering patterns suggest similar circulation dynamics, though the slow mass diffusion process can occur after layering.

Modification in the revised manuscript.

We added a discussion of the Schmidt number on P. 6 of the revised manuscript: “The Schmidt number $Sc = \nu/\kappa_s \approx 300$ (κ_s is the mass diffusivity of the salt), indicating the momentum diffusion is far faster than the salt diffusion during both the injection and layering (if any) processes. The Schmidt number for the milk and espresso system is approximately 10^4 , and so is $\gg 1$ as the model salt and water system, in which momentum diffusion dominates.”

We referenced the double diffusion convection phenomenon in the ocean and lakes on P. 7 of the revised manuscript and cited related references: “We postulate that this final layered state is obtained due to the *double-diffusive* convection, which is well-known in layer formation in open water systems such as oceans or lakes.” We also referenced the double diffusion convection phenomenon in open water system in the first paragraph of the introduction.

3) Referee: DDC is controlled by TWO Rayleigh numbers, namely the one for the temperature and the one for the concentration field. They can be either positive (destabilizing) or negative (stabilizing). Both Rayleigh numbers should be given.

Our response.

Typically, double diffusive convection is controlled by the Rayleigh number for the temperature and the Rayleigh number for the concentration field, where the temperature and concentration gradient are both vertical, i.e., parallel to gravity. In this condition, the two Rayleigh numbers can be either positive (destabilizing) or negative (stabilizing). In our model experiments, in contrast, according to the way we have conducted the experiment, the temperature gradient (horizontal) is orthogonal to the concentration gradient (vertical), and so the temperature gradient is always destabilizing and the concentration gradient is always stabilizing. According to Ref. [15] (by C. Chen, D. Briggs and R. Wirtz, 1971), in the configuration of our experiments, a different Rayleigh number, which combines the horizontal temperature gradient and vertical concentration gradient, controls the double diffusive convection and layering. We reported this Rayleigh number in our paper.

Modification in the revised manuscript.

To better recognize the distinction between the Rayleigh number defined in the present manuscript and the typical definition of Rayleigh number, we modified the sentence describing the Rayleigh number on P. 9 by stressing the directions of the gradients: “The competition between the thermal cooling and the salinity gradient in this double diffusive problem, where the temperature gradient (horizontal) is orthogonal to the salinity gradient (vertical), is characterized by the Rayleigh number, $Ra = \frac{g\alpha\Delta T}{\nu\kappa} \left[\frac{\alpha\Delta T}{(-d\rho/dz)/\rho_w} \right]^3$.”

4) Referee: One reason why I worry on the Schmidt number are the numerical simulations by COMSOL. I somehow doubt that COMSOL can handle so large Schmidt numbers as it would require excellent grid resolution for the concentration field (which has much smaller diffusivity). Anyhow, the authors say that they have validated their code and found a good match, but clearly this remains some major concern. Perhaps it still works as the Rayleigh numbers are not high.

Our response.

We agree with the referee that a highly refined mesh is required in order to capture the details of the flow phenomenon. However, we think such refinement goes beyond the scope of current numerical study which aims to show, in a qualitative sense, the formation of layers and the the co-rotating vortices confined in these layers to supplement our experimental observations. To check our simulation results, we validated our implementation against a 2D numerical study (Ref. [30], see Fig. 1 below) and a 3D axisymmetric one (Ref. [29], see Fig. 2 below) by comparing the temperature and concentration contours. Fig. 1 and 2 show that our numerical results mostly agree with the published results but not precisely. We argue that the discrepancy might stem from the relatively low grid resolution used by the two studies carried out in the 1990s. For example in Ref. [30], a finite element method using linear elements was adopted, with a grid resolution of 31×41 nodal points.

Besides the validation shown, we also carried out a simulation with a refined mesh consisting of approximately 30000 quadratic elements compared to 6000 elements of the simulation presented originally. We focus on the non-dimensional salinity distribution σ at the middle radial position $r = R/2$ along the depth direction z and compare the data of the two simulations (see Fig. 3 below). We do not observe a qualitative difference between the two, at least in the regions where the layers emerge ($z > 1$). Therefore, we believe that the original simulation has captured qualitatively the main physical features, as also confirmed by the reported agreement with the flow field from PIV measurements.

5) Referee: The numerical simulations are 2D in the sense that (r,z) cylindrical coordinates are used. Why so? Why not simply 2D in the (x,z) plane? Note that there are considerable difference between 2D and 3D RB convection. The best of course would be 3D DDC, but indeed I can imagine that COMSOL then fully fails.

Our response.

The experiments presented in the manuscript are performed in a cylindrical tank containing a mixture that cools at room temperature, so the resulting circulation

flow is expected to be axisymmetric in the cross-sectional plane of the tank. Therefore, 2D cylindrical coordinates, i.e., (r,z) would be the best representation for the experiments in the numerical simulations. Our experimental observations support the interpretation that the results are, at least to a good approximation, axisymmetric. Of course, it would have been sufficient to do a 2D simulation to capture the ideas; however, our goal was to convince readers unfamiliar with these physical systems that our simulations represented our experimental system.

6) Referee: Figure 2b: The authors may consider to switch the axis and plot z vs ρ , as more common in DDC. But it is of course up to them.

Our response.

We appreciate this suggestion. We switched the axes in the plot to a more common format.

Modification in the revised manuscript.

We switched the axes in the plot (figure 2b) and adjusted related notations in the manuscript.

FIGURE 1. Validation against Ref. [30] of a 2D double diffusive convection simulation where temperature and concentration gradients are imposed in the horizontal direction. The non-dimensional numbers $Pr = 1$, $Ra_T = 10^5$ and $Le = 2$ are fixed. Two buoyancy ratios $N = 0.8$ and 1.3 are tested. Top: the data extracted from Figure 2 of Ref. [30]. Bottom: our numerical simulation.

FIGURE 2. Validation against Ref. [29] of a 3D axisymmetric double diffusive convection simulation where temperature and concentration gradients are imposed in radial direction. The non-dimensional numbers $Pr = 7$, $Ra_T = 5 \times 10^4$ and $Le = 5$ are fixed. Two buoyancy ratios $N = -10$ and -2 are tested. Top: the data extracted from Figure 3 of Ref. [29]. Bottom: our numerical simulation.

FIGURE 3. Comparison of the salinity distribution at three time instants of two simulations. The first (blue solid curves) uses approximately 6000 quadratic elements and the second (black dashed curves) 30000 elements.

ANSWERS TO REFEREE 2

Referee: This paper presents an experimental and numerical examination of an elegant method for producing layering using a simple double diffusive fluid system. Layering is of relevance to a number of industrial and environmental processes and so the identification of techniques to simply and repeatably reproduce layering are worthy of publication in their own right. The study presents some details of the physics behind the layering process which provides insight into the underlying processes involved. The study then goes on to demonstrate application of their technique for producing layers by application to soft matter gels which are of wide use in modern society.

The paper is very well written and well presented. I would not be unhappy to see the paper published as is but I have four comments which the authors may like to consider:

Our response.

We thank the referee for his/her valuable suggestions. We are pleased to read “The paper is very well written and well presented. I would not be unhappy to see the paper published.” Below we address the suggestions of the referee and include the modifications to the manuscript wherever applicable.

1) Referee: The layers in the relevant images are visible when I view on-screen but in my printed version they are impossible to detect. I wonder if any more can be done, e.g. some carefully targeted stretching of the light intensity values, to the images to accentuate the visible appearance (or otherwise) of the layers in the images?

Our response.

We thank the referee for this suggestion. We tried to enhance the contrast of the images, but there is very little improvement in visualizing the layers. Thus, we choose to maintain our original experimental images. The quality of the printed images are strongly dependent on the printers. As Nature Communication is an on-line only journal, the printed image quality may not be a significant issue, and we imagine that many readers will simply read from computer screens, where the images are fine, as the referee notes.

2) Referee: I dislike the notion that the formation of layers is dependant on solely the velocity of injection. The appearance of layers must in general result from the initial stratification established and the rate of heat transfer through the walls of the vessel. The initial stratification must depend on the velocity of the injection but only in combination with the opposing effects of buoyancy, which typically for these fountain-like flows is expressed via a Froude number. Indeed, considerations in this vein provide that for sufficiently forced injections, the volume of 'mixed fluid' would be expected to vary linearly with injection volume (as is described to have been observed on p7) since the bulk mixing/entrainment by sufficiently forced fountain is linear in the Froude number (see for example Burrige HC, Hunt GR, 2016, Entrainment by turbulent fountains, Journal of Fluid Mechanics). I accept that you did not vary the Froude number except by varying the velocity but I think you should at least comment that simple considerations of the physics allows one to identify that the mixing process must be controlled by the Froude number of the injection. Personally I would present the horizontal axis in figure 3b in terms of a Froude number as the information regarding the critical velocity, only relevant to your particular set-up, is in figure 3a already ? information regarding a critical Froude number may be applied to experiments of different initial density differences at the very least. Finally on this point, I would like to see a comment that one might expect the height of the layers to be determined by the initial density gradient and rate of change of fluid density due to heat transfer through the vessels side-walls, i.e. the characteristic distance 'cool' fluid falls before being neutrally buoyant in the stratified 'mixed' fluid. I would imagine that the layers only form when the height scale of the layers is large compared with a length associated with diffusion within the fluid but small compared with the height of the initial mixed region.

Our response.

We thank the referee for this remark. We certainly agree with the referee that the initial stratification is not solely dependent on the velocity of injection and we indicated more general considerations in our manuscript. The Froude number and the injection volume also play roles in establishing the initial stratification. In addition, the Reynolds number is also important in our model experiments since the injection Reynolds numbers fall in a transition regime between laminar and turbulent flows. To highlight the effect of the variable which is the simplest to control in a pouring experiment, we chose the velocity of injection. Identifying the appropriate non-dimensional parameters for characterizing the mixing for the range of Froude and Reynolds number of interest is beyond the scope of our manuscript, and is a subject

of on-going research, as we have done a large number of experiments on this topic, which we hope to report in the future in the fluid mechanics literature.

The expected height (or thickness) of a layer is determined by the initial density gradient and the temperature gradient. The length scale of the layers is approximately $(\alpha\Delta T)/[(-d\rho/dz)/\rho_w]$, as reported in Ref. [15] (C. Chen, D. Briggs and R. Wirtz, 1971).

Modification in the revised manuscript.

As suggested by the referee, we added the remark “Consideration of the dynamics in a proper dimensionless framework requires an analysis with at least both the Froude and Reynolds numbers, which is a topic of on-going research.” on P. 9 of the revised manuscript.

We added a comment on the expected height of the layers on P. 9 of the revised manuscript: “In addition, when layering occurs, the expected length scale (thickness) of a layer is approximately $(\alpha\Delta T)/[(-d\rho/dz)/\rho_w]$.”

3) Referee: 3) The jump, on page 8, from the discussion of critical Rayleigh numbers to the application of the technique to soft matter gels is far too stark and detracts from this interesting final application. There need to be a(some) more introductory sentence(s) beginning the the section on soft matter gels.

Our response.

We thank the referee for this remark. We revised the manuscript to address this suggestion.

Modification in the revised manuscript.

On P. 9-10, we added “The double-diffusive convection and the formation of the layers are simply controlled by the thermal and salinity gradients in the fluid systems discussed above, which implies no conceptual restriction for applying this principle to more complex fluid systems, such as thermally established soft gels.”

4) Referee: My only concern that truly requires addressing prior to publication, is that the image presented in figure 4a bears little evidence of layering (both when viewed on-screen and in-print). For example there is no indication that between any two consecutive dashed horizontal lines drawn on figure 4a the gel is of a constant colour ? quite the opposite, if one ignores the horizontal dashed lines then my eyes (at least) identify a relatively continuous variation in colour over the entire region in which the colour is varying. If the variation is continuous then one would still expect the material properties to vary with height, which you demonstrate but this

is not evidence of genuine layering. The real question is: if you divided the entire material in figure 4a into sections, each of a height equal to half the distance between your dashed lines, then could one identify a significant and systematic variation in the material properties based on whether two consecutive samples lay either within the same two dashed horizontal lines or were separated by a dashed horizontal line? Addressing this point is key to being able to claim genuine layering, rather than a continuous variation, in the material properties of the gel formed. If more robust evidence cannot be produced then at the very least this points needs to clearly discussed and acknowledged in the text.

Our response.

We thank the referee for this suggestion. One of the other referees also gave us a similar suggestion.

We agree that it is important to demonstrate the layering in the gel. The agarose gel, however, is semi-transparent and thus the layers appear blurred. The layers are clearer when observed by eye, and we have tried to improve the quality of images. Nevertheless, we repeated the experiment with the same recipe and were able to get improved images of layering with clear steps in the image-processed signals, which we have now reported in Supplementary Discussion and Supplementary Fig. 6.

Modification in the revised manuscript.

We added a new figure and a paragraph about the layered gel in Supplementary Discussion and Supplementary Fig. 6. We referenced the SI in the manuscript to let readers know about this clarification of layers, as “To further demonstrate the presence of the horizontal layers in the gel, we performed experiments with the same recipe but measured the light intensities in the digital images captured from the gel rather than the elastic properties, see Supplementary Fig. 6 and Supplementary Discussion.”

To clarify the layering and the solidification process, we also added a sentence in the last paragraph of the manuscript: “When cooling, horizontal layers are first formed in the agarose solution, which is subsequently solidified to a layered gel below the gelation temperature.”

ANSWERS TO REFEREE 3

Referee: The manuscript describes the formation of density layers in a cooling mixture of brine and dyed water using a jet of opposing buoyancy. This phenomenon is also observed while injecting coffee in milk and is rationalized as a thermosolutal convection process where the diffusivities of heat and solute produce density differences that drive fluid motion. The authors show in their model experiments and numerical simulations that the formation of layers is due to recirculating flow cells, which locally mix liquids. A critical velocity for the onset of fluid layering is reported and a similar approach is employed for generating layered soft materials. The manuscript presents interesting results but requires a few clarifications, including:

Our response.

We thank the referee for his/her valuable comments. We are pleased to read “The manuscript presents interesting results...” Below we address the comments and suggestions of the referee and include the modifications to the manuscript whenever applicable.

1) Referee: The formation of horizontal layers is evidenced though analysis of grey-scale intensity profiles as explained in the SI. Data displayed in Fig. 2(b) of the main manuscript, however, do not provide compelling evidence of fluid layering as density steps only appear in a small portion of the graph and for one case. A revised figure clearly showing multiple step-wise intensity profiles would be more convincing. Such clarification is also necessary for Fig. 2 of the SI as intensity layers also appear in the images associated with low injection velocities (i.e., $U = 0.07$ and 0.20 m/s) where no layering is reported. In addition, a few quantitative remarks about the widths of the layers observed in both experiments and simulations would allow for a better connection between these approaches.

Our response.

The layering as density steps only appear in a portion of the mixture. For example see Fig. 2(b) where complete layering only occurs when the density (concentration) gradient is relatively small, say, the top part of the mixture. As for the bottom part of the mixture, the density gradient is relatively high and this stabilizing effect weakens the double diffusive convection. As a result, clear steps cannot be observed around the bottom of the mixture where the density gradient is relatively high. Moreover,

Fig. 2(b) aims to show that the layering and the consequent appearance of steps in the gray-scale intensity analysis occur only at higher velocity and only for one of the cases.

We revised Fig. 2 of the SI to demonstrate the presence of layers. The image intensities, to our understanding, provide similar evidence of layering as density profile. As a result, we choose to plot the density profile to show the layers because density profile is more common and more related to the contents. We plotted the density versus the position z , see Fig. 2(b) in SI. Vertical steps are observed at $U = 0.23, 0.40$ and 0.67 m/s, indicating layers formed. The density profiles at $U = 0.07$ and 0.20 m/s are relatively smooth and no steps are observed. These results indicate that no layer is formed at low injection velocity.

The expected height (or thickness) of the layers is determined by the initial density gradient and the temperature gradient. The length scale of a layer is approximately $(\alpha\Delta T)/[(-d\rho/dz)/\rho_w]$, as reported in Ref. [15] (C. Chen, D. Briggs and R. Wirtz, 1971).

Modification in the revised manuscript.

We added the density plot showing layering in Fig. 2(b) of the SI. We revised the related captions and the texts.

We added a comment on the expected height of the layers on P. 9 of the revised manuscript: “In addition, when layering occurs, the expected length scale (thickness) of a layer is approximately $(\alpha\Delta T)/[(-d\rho/dz)/\rho_w]$.”

2) Referee: Although the presence of a concentration gradient is apparent during the formation of the soft material in Fig. 4, no evidence is provided for the formation of a layered structure with a step-wise variation of the Young modulus E . A clear quantitative distinction should be made between a smooth and a step-wise variation of material properties. Providing the step-wise evolution of E as a function of z measured from consecutive thin slices would better substantiate the formation of layers resulting from double-diffusive convection.

Our response.

We thank the referee for this suggestion. One of the other referees also gave us a similar suggestion.

We agree that it is important to demonstrate the layering in the gel. Continuously plotting the Young’s modulus E versus z and showing a step-wise variation is very useful and is a subject of on-going research. In this paper, we use the light intensity measurement to prove the layers in the gel.

We repeated the experiment with the same recipe and were able to get improved images of layering with clear steps in the image-processed signals, which we have now reported in Supplementary Discussion and Supplementary Fig. 6.

Modification in the revised manuscript.

We added a new figure and a paragraph about the layered gel in Supplementary Discussion and Supplementary Fig. 6. We referenced the SI in the manuscript to let readers know about this clarification of layers, as “To further demonstrate the presence of the horizontal layers in the gel, we performed experiments with the same recipe but measured the light intensities in the digital images captured from the gel rather than the elastic properties, see Supplementary Fig. 6 and Supplementary Discussion.”

To clarify the layering and the solidification process, we also added a sentence in the last paragraph of the manuscript: “When cooling, horizontal layers are first formed in the agarose solution, which is subsequently solidified to a layered gel below the gelation temperature.”

3) Referee: It has been reported that double-diffusive phenomena can also occur in the absence of heat when two solutes, such as salt and sugar, are present [Ref. 14, page 47]. In the reported experiments, salt and dye also constitute two different solutes and it would be useful to clarify whether additional experiments with variations of salt and dye concentrations along with various temperature differences were conducted. In general, more details about the influence of temperature would strengthen the manuscript.

Our response.

We have done the experiments that study the effects of temperature. In our present system, no circulation flow is generated according to our PIV results in the absence of heat transfer. Every particle is static. So the thermal gradient in our experiments is necessary.

Modification in the revised manuscript.

On P. 6 of the revised manuscript we added the sentence “Layers can be only observed in either the milk and espresso or the salt and water mixture when the initial temperature is different from room temperature.”

4) Referee: The schematic of the model experiment presented in Fig. 1(b) suggests at first that step (2) and (3) are conducted at the same time.

Our response.

Steps (2) and (3) are conducted at the same time. Dyed water is injected into the salt solution at the same time when the whole container is cooling at room temperature. However, the injection just lasts for tens of seconds, while the cooling lasts for hours. So we note the injection as step (2) and cooling as step (3).

Modification in the revised manuscript.

To clarify the steps, in the caption of Fig. 1b in the revised manuscript, we added a sentence “The cooling begins as soon as the injection starts and lasts for at least several minutes.”

Reviewers' Comments:

Reviewer #1:

Remarks to the Author:

Nice job. The paper should now be accept as is. Very nice and inspiring work and a pleasure to review.

Reviewer #2:

Remarks to the Author:

The revised manuscript is now suitable for publication. The wording regarding the layering in the agrose gel is now very careful and supplementary figure 6 is reassuring that there is at least some evidence of layering. I support publication of this manuscript.

Reviewer #3:

Remarks to the Author:

Previous comments were satisfactorily addressed, I recommend publication.